# Where do People Interact in High-Rise Apartment Buildings? Exploring the Influence of Personal and Neighborhood Characteristics

**DOI:** 10.3390/ijerph17134619

**Published:** 2020-06-27

**Authors:** Linh Nguyen, Pauline van den Berg, Astrid Kemperman, Masi Mohammadi

**Affiliations:** Department of the Built Environment, Eindhoven University of Technology, 5600MB Eindhoven, The Netherlands; P.E.W.v.d.Berg@tue.nl (P.v.d.B.); a.d.a.m.kemperman@tue.nl (A.K.); M.Mohammadi@tue.nl (M.M.)

**Keywords:** high-rise apartment buildings, social interaction, personal characteristics, neighborhood characteristics, spaces for social interaction

## Abstract

Early studies conclude that high-rise apartment buildings present challenges for people’s quality of life, resulting in social isolation, social annoyance and anonymity for residents. Nevertheless, empirical research into factors supporting social interaction in high-rise apartment buildings is still scarce. This study aims to investigate how often and where people in high-rise neighborhoods interact, and how this is affected by personal and neighborhoods characteristics. A mixture of both quantitative and qualitative methods was used including social interaction diaries and questionnaires among 274 residents, in-depth interviews with 45 residents and objective measurement of the physical environments in four high-rise apartment buildings for low in-income people in Hanoi, Vietnam. Results demonstrate that social interaction is influenced by a number of personal and neighborhood characteristics. Furthermore, most social interactions—also gathering and accompanying playing children—take place in the circulation areas of the apartment buildings. However, the use of these spaces for different purposes of interaction is found to have negative impact on people’s privacy, the feeling of safety and cleanliness of the shared spaces. The findings of this study provide information for planners and designers on how to design and improve high-rise apartment buildings that support social interaction.

## 1. Introduction

Many studies have focused on social aspects of residential areas in modern cities. These studies have pointed out that social interaction is a very important factor of a community. Daily social interaction plays an important role in the richness and liveliness of social life [1,2] and it can improve individuals’ well-being and happiness [3,4]. Regarding social interaction in the neighborhood, the built environment can play a key role in inviting residents to come outside their private territories and as a result, support communication among residents.

As urban populations continue to grow and space is limited, vertical buildings play an important role in the city development. However, specific studies on high-rise apartment buildings conclude that they ignore the needs and lifestyle habits of their residents [5,6]. The (often uniform) buildings have created different problems, such as: behavior problems, helplessness, poor social relations, mental health and hindered child development [7]. Crowding in high-rise buildings makes it difficult to control social interaction [8] which may isolate people from each other [9]. Therefore, people living in high-rise buildings are considered to have weaker social relationships, either with their fellow neighbors and/or with the outsiders [7]. Kearns et al. (2012) [10] compare different social issues (e.g., social cohesion, social interaction and social support) between high-rise buildings and other housing typologies and conclude that high-rise living has a high negative impact on residents. Therefore, it is essential to provide opportunities and places for social interaction in high-rise, high-density residential areas to build community cohesion amongst residents.

Although, past research has studied the relationship between high-rise neighborhoods and social interaction [3,10,11,12,13,14,15,16,17,18,19], little attention has been paid to factors supporting social interaction in this living environment. These studies have focused on the importance of spatial, social and environmental aspects of the neighborhoods but not provided a detailed assessment of the relationship between personal characteristics and the characteristics of their living environments and the impact they have on social interaction. In addition, most of the recent studies on apartment buildings in Vietnam focused on the physical characteristics of the living environments including sustainable apartment design (energy efficiency, design and spatial structure) [20,21,22,23], apartment living and fire safety [24,25], housing satisfaction in apartment buildings and its correlates [26,27,28,29,30]. To the best of the authors’ knowledge, this study is the first empirical research to investigate factors that support social interaction among high-rise residents in Hanoi.

This study makes a contribution in that it aims to investigate the influence of personal and neighborhood characteristics on social interaction among residents of high-rise apartment buildings. Specifically, the research question is: How often and where do people in high-rise neighborhoods interact, and how is this affected by personal and neighborhoods characteristics? Mixed methods were used to answer this question, including social interaction diaries, questionnaires, in-depth interviews and objective measurement of the neighborhoods. Cross-tabs, analyses of variance and chi-squared automatic interaction detection (CHAID) analysis will be used to explore the influences of personal and neighborhood characteristics on social interaction. Data collected by in-depth interviews will be added to underpin the findings.

The next section will first discuss the concept of social interaction and its importance for people’s well-being. Subsequently, factors that have been found to influence neighborhood social interaction are discussed. Section 3 presents the data collection procedure, followed by the analyses and results in Section 4. The final section presents the conclusions and discusses the implications of the findings.

## 2. Literature

### 2.1. Neighborhood Social Interaction

Neighborhood social interaction has been receiving increasing attention in recent years. Although neighborhood ties are considered to be weak compared to the strong relations between friends and relatives [31], social interaction in the local neighborhood is recognized as an important indicator to enhance quality of social life and the feeling of place attachment [32]. Scholars agree that neighborhood contacts can improve social cohesion and sense of community [13,33,34], life satisfaction [35,36] and promote social well-being [32,37]. Interacting with neighbors such as exchanging interests or small conversations could increase people’s happiness, health and well-being [4].

Social interaction provides residents with information about their fellow neighbors and the social structure of the community, and therefore supports the primary process in developing neighborhood communities [38]. This in turn builds friendship patterns and social connection, and creates common rules of community [39]. With a certain level of social interaction, people can have the feeling of being socially integrated and this could decrease the feeling of loneliness [40]. Specifically, Chile et al. (2014) [41] indicate that a low level of social contact with other members of a community is the main factor that creates social isolation for inner-city high-rise apartment residents.

### 2.2. Personal Characteristics Influencing Social Interaction

Personal characteristics are considered important indicators for influencing neighborhood social interaction. Age is found to have significant influence on the number of social interactions. Older people are less likely to interact with neighbors compared to younger age residents [42,43]. Home ownership [44] and length of residence [45,46] are positive indicators that increase social interaction between neighbors. The longer someone lives in the neighborhood, the stronger the connection they have to their community [45]. Households with children tend to have a richer neighborhood relationship and more neighbors in their networks [38,47]. Higher-income people often have more access to resources, so they tend to have more contacts outside their neighborhood and therefore less need to communicate with fellow neighbors [48]. In contrast, the lower-income group has a greater tendency to interact with geographically close neighbors than their relatives who are far away (Jo and Jo 1992) [49].

### 2.3. Spatial and Physical Characteristics of the Neighborhood Influencing Social Interaction

Interaction with neighbors is related to aspects of the neighborhood. Whether or not people are willing to spend their time to interact with neighbors depends on the experience that they have in the environment [3,50,51]. Williams (2005) [52] states that social interactions are stimulated when residents have opportunities for contact, live in proximity to each other and have appropriate spaces for social interaction. Furthermore, neighborhoods which can provide shared pathways from private units to different activity spaces can increase opportunities for social interaction [50].

Previous studies show mixed results regarding proximity. Proximity attributes such as the number of dwellings in a block, the number of dwellings sharing one entrance, the constellation of houses or apartments are assumed as important predictors for the success of social interaction [50,51]. Glaeser and Sacerdote (2000) [45] conclude that big apartment buildings reduce the distance among residents and increase the distance between residents and the streets. Therefore, residents that are more physically proximate will be more connected with their neighbors [4]. Delmelle et al. (2013) [4] agree with this conclusion by indicating that a denser neighborhood increases face-to-face and spontaneous interactions between residents. However, there are arguments that people who live in single detached homes might communicate more with neighbors due to the direct connection with shared open space outside their homes [45]. Altman’s privacy theory (1977) [53] emphasizes the negative impacts of density or proximity on neighboring. It indicates that if a certain level of privacy cannot be controlled or involuntarily encounters take place, social cohesion can be decreased within the neighborhood. Furthermore, Thompson (2018) [15] indicates that the lack of private space in high density residential areas reduces social contact among residents.

The provision of appropriate space for social interaction in the neighborhood is very important in encouraging social interaction. Residential neighborhoods are potential places for social contact and developing social relations [47], thus it is essential to consider the design of a neighborhood in such a way that it encourages residents to get out of their private home and out into the social space [54]. Neighborhood social spaces are considered to be good when they are formally shared and accessed by familiar neighborhood people, where they can have casual interactions, shared responsibility and expand their social networks [47]. Previous studies mention space layouts and the availability of facilities in communal spaces [16,50] as important factors of the living environment in fostering social activities and informal social interaction. The diversity, quality, accessibility and visibility of communal spaces can be considered as key design variables influencing social interaction [52].

High-rise apartment buildings have received plenty of criticism from researchers in different disciplines. Gifford (2007) [7] concludes that high-rise living environments have both advantages and disadvantages as they provide greater privacy and limit unwanted social interaction, and at the same time they reduce intimate social interaction and less caring between residents. Moreover, Evans et al. (2003) [8] state that the lack of shared spaces within high-rise apartment buildings such as lobbies and lounges, which are considered as appropriate spaces to maintain social networks could probably lead to social isolation. Furthermore, Modi (2014) [12] points out that communal space in high-rise apartments normally does not accommodate the daily activities of residents.

Literature confirms the interrelation between social interaction and personal characteristics and the characteristics of the living environment. In addition, previous studies conclude that high-rise living environments can result in social isolation, social annoyance and anonymity for their residents. However, research on factors supporting social interaction in high-rise living environment is still scarce and requires more empirical evidence.

## 3. Methods

A mixed methods approach is used to answer the research question including (1) a questionnaire to measure personal characteristics and the experience with and perception of the apartment building and neighborhood environment, (2) an interaction diary for one day to record the social interactions people have with others in their apartment building and neighborhood, at what specific location and for which purpose, (3) qualitative interviews to acquire in-depth insight in residents’ experience with their social and physical living environment and their social behavior in their neighborhood. Moreover, (4) physical characteristics of the neighborhood are objectively measured.

### 3.1. Questionnaire

A questionnaire was designed to collect information on personal characteristics including age, gender, employment status, length of residence, background, education status, household size and composition and home ownership.

### 3.2. Interaction Diary

In order to understand the social interaction in the neighborhood, an interaction diary was developed following the social interaction diary method proposed by Van den Berg et al. (2015) [55]. This diary asked people to record the number of social interactions they have with neighbors (for one day) and which spaces are used for different activities. There must be a face to face (direct) contact for social interaction between two or among more people including conversations such as a greeting or talking and doing something together with someone else. Note that interactions by phone or internet such as a phone call, an email, a text message, were not taken into account.

### 3.3. In-depth Interview

In-depth interviews were held with 45 residents of the various apartment buildings to acquire their background and experiences with their social and physical living environments. An interactive, flexible approach was used during the interviews. Questions were adapted depending on the respondents’ answers to gain more in-depth insight. Open-ended questions allowed respondents to use their own words and express their opinion about a situation. Thus, the interviews were more like a guided conservation than a strict structured interview.

### 3.4. Objective Measurement of the Physical Environments

Previous research [56,57] investigates the walkable area in a neighborhood and conclude that 500 m is seen as a representative distance people are willing to walk from their home. Therefore, this study focusses on the residential neighborhood within the building and the surrounding area within a radius of 500 m.

An objective measurement was conducted to measure the physical characteristics of the neighborhoods. The site plans were drawn including the location of the block(s) and the surrounding areas including green/open spaces, sport facilities, communal spaces, the location of the block(s) and the surrounding areas including green/open spaces, sport facilities, communal spaces.

## 4. Data Collection

The data was collected between July and September 2019 in four different high-rise apartment buildings and neighborhoods for low-income people in Hanoi, the capital city of Vietnam.

Apartments for low-income people are in great demand in Vietnam’s large cities [58]. In Vietnam, housing for low-income people is recognized as an important component to complement the development of higher productivity jobs in cities, supporting 40% of the population, as well as addressing the national targets for the ongoing rapid urbanization [59]. There are two types of housing for low-income people: social and low-cost housing. Only employees of public institutions (e.g., teachers, soldiers, students, labor workers) who have monthly income less than 350 euro and do not have any house under their homeownership can purchase social housing. Conversely, low-cost housing can be purchased by everyone from private and public institutions. Hanoi is the capital city of Vietnam which is located in the northern region of the country. Its administrative boundaries have expanded since 2008 when the city annexed a neighboring rural province, increasing its size threefold. This expansion has made it the second largest city by population, with approximately eight million people as of 2019 and an area of 3359 km^2^. To date, more than 150 new urban areas have been developed, 50 of those have been implemented and the remainder are in different phases of planning and construction [60]. Most of the projects are located at a distance of 7–12 km from the city center. Ha Dong, the selected district of this study has been the fastest growing district in Hanoi. It is located on the development axis in the southwest of the city which is approximately 10 km from the city center, and is expected to become a multifunctional sub-center that will reduce pressure on the growing population in the city center. In this district, two social high-rise apartment projects and two low-cost apartment projects were selected. These buildings and neighborhoods were chosen based on their perceived cohesion despite conditions that make it difficult to develop neighborhood ties: large scale, recent development (all buildings and complexes were occupied after 2010), lack of shared open spaces. Location of the four neighborhoods is presented in Figure 1.

To recruit respondents for participating in the questionnaires and interaction diaries, a personal approach was used. First, permission was sought with the management board of the neighborhood/building association. Subsequently, residents were visited at home and asked if they were willing to participate in the questionnaire and interaction diary. The visits took place at varying times of day, also in the evening, to make sure that also working people were included in the sample. Out of 400 approached residents, 274 (70%) people agreed to participate and have completed the questionnaire and interaction diary (Kien Hung: 73; SDU: 106; Thanh Ha: 80; Dai Thanh: 15).

Respondents for the in-depth interviews were approached during the distribution of the questionnaires and interaction diaries. Past research indicates that sample size in interview studies is often justified by interviewing participants until reaching data saturation. Francis et al. (2010) [61] and Guest et al. (2006) [62] conclude that about 12 interviews are needed to reach saturation. This study was conducted in four different neighborhoods; therefore, 60 people were first proposed for the in-depth interviews, of which 45 respondents were interviewed until limited new information was obtained. The numbers of residents that participated in the interviews were 22, 10, 10 and 3 from Kien Hung, Thanh Ha, SDU and Dai Thanh, respectively. Before the in-depth interviews were conducted, residents were informed about the purpose of the study and given assurance about ethical principles for instance anonymity and confidentially. On average, the interviews lasted 30–60 min.

Objective environmental data including housing typology, year of occupation, number of blocks, number of apartments per block and number of stories per blocks were available through the building management. Site plans of four neighborhoods were drawn based on google maps and later were double checked by on-site observation and sketching.

## 5. Analyses and Results

The aim of this study is to investigate how often and where people in high-rise neighborhoods interact, and how this is affected by personal and neighborhoods characteristics. In this section, descriptive statistics will be presented followed by bivariate analyses (cross-tabs and analyses of variance) and CHAID analysis. In addition, to verify the findings from social interaction diaries and questionnaires, results from the in-depth interviews will be discussed.

### 5.1. Socio-demographic Characteristics of Respondents

Table 1 shows the personal characteristics of the 274 respondents who participated in the social interaction diaries and questionnaires in the four neighborhoods. Only respondents over 18 years old are included in this study. About 50% of respondents are aged 35 to 54, followed by 40% aged 18 to 34. There is only a small percentage of respondents over 55 years old. Compared to the population of Hanoi, the sample contains a lower percentage of the oldest group (55 and older) and a higher percentage of the two younger age categories. It reflects the age composition of residents of high-rise apartment buildings. There are slightly more female respondents than males, and this distribution is associated with the total population. This might be due to the fact that women are more likely to be open for a conversation than men, and they might be more often at home. There are substantially more employed people (employed and free-lancers) than unemployed which corresponds with the population numbers. Nevertheless, unemployed people are somewhat overrepresented in the sample. An explanation for this might be that the four neighborhoods are high-rise apartment buildings for low-income people. Overall, the sample does not completely represent the entire population of Hanoi, however it does reflect the socio-demographic characteristics of high-rise apartment buildings.

### 5.2. Description of Neighborhoods

Table 2 presents the environmental characteristics of the four neighborhoods. SDU is the only single high-rise building whereas other neighborhoods include multiple high-rise buildings. The earliest occupied neighborhood is Kien Hung (2012) whereas Thanh Ha is the latest one (2018). Residents of Dai Thanh moved into the neighborhood in 2014 which is one year before the SDU residents (2015).

As can be seen in the site plan, Kien Hung has no official green/open spaces or sport facility within the distance of 500 m. There is an open market and a self-built vegetable garden in the nearby neighborhood which are also a platform for residents to socialize. A common room (meeting room) is provided in each building.

There is a lake/park in the distance of 100 m in Thanh Ha. The three buildings share one common room and one front court with few benches for the whole population. No sport facility is present within 500 m.

SDU is located in a high-density residential area of the city and is directly connected to the main street. There are no yard, playground or any open space surrounding the building. Lakes and open spaces are in a distance of 500 m of busy streets without a linking walkway to the sites. This is the only building which provides communal spaces for children to play and study, and spaces for residents to do exercise although they are small. There is a sport facility in a distance of 500 m.

Dai Thanh has no green open space/courtyard within the distance of 500 m. There is a walkway around the buildings with benches. Each block has one common room (but it is rarely used). Spaces in the ground floors are almost completely occupied by private services such as tea shop, coffee, beauty salon and even paid play-spaces with facilities for children.

In summary, SDU is the only neighborhood which provides different communal spaces for different purposes within the building; they are small (but frequently used by residents). These spaces in other neighborhoods are rarely used or are transformed to use for different purposes. SDU is also the only neighborhood which has no open shared space or courtyard around the building; other neighborhoods provide small yards or walkways with benches. A lake and park are present in Thanh Ha and SDU in the distance of 100 and 750 m, respectively. There are sport facilities within a distances of 500 m in SDU and Dai Thanh, whereas there are no such facilities in Kien Hung and Thanh Ha.

### 5.3. Social Interactions

In total, 1095 social interactions were recorded in one-day interaction diaries by 274 respondents of the four neighborhoods. People had a minimum of one interaction and a maximum of ten interactions with neighbors during the day, and on average each respondent had four interactions with neighbors. Figure 2 shows the distribution of the number of social interactions during one day. Two hundred and ten respondents (77%) recorded their social interaction diaries during a weekday whereas 64 respondents (23%) did so during a weekend. Not surprisingly, residents record significantly more interactions in the weekend (on average 4.58 interactions per day) than the weekdays (on average 3.82 interactions per day).

Although all respondents had at least one social interaction on the day of the interaction diary, many of them indicated that they do not have much contact with their fellow residents and would prefer to have more social interaction.
People live in a close distance but not many people have contact with neighbors.(73_man_owner_SDU)

During weekdays, social interactions were mostly in forms of small talk and greeting whereas in the weekend, people spent time for longer interactions such as gathering and visiting neighbors. Therefore, higher numbers of social interactions were recorded on weekend days.
During the day on weekdays most people just say Hi or have a quick chat with neighbors on the way to go to work or get back home.(61_man_owner_Kien Hung)
During the weekend, I sometimes visit neighbors or invite some men to my home for a drink and have a talk.(49_man_owner_SDU)

### 5.4. Main Purposes, Spaces Used and Importance of Social Interaction in the Neighborhoods

Table 3 shows the distribution of the recorded social interactions across locations, main purposes and the importance of social interaction. The findings show clearly that most social interactions among respondents are greeting/short messages (45.2%) and long talks/chats (33%). Joint activities and accompanying kids for co-playing follow at about 11%. SDU has the highest percentage of greeting/short messages and the lowest percentage of a long talk/chat across neighborhoods. It might be due to the fact that SDU has no open shared spaces (yard/walkway and benches for sitting) around the building for people to stay for a longer conversation. This may also be an explanation for the lower share of joint activities and accompanying kids for co-playing in this neighborhood. Thanh Ha has the smallest share of joint activities and highest share of accompanying kids for co-playing. The contrary goes for Dai Thanh, which has the lowest share of social interactions while accompanying children. This is probably related to the duration of stay in these two neighborhoods. The presence of small yards/walkway (with benches) around the buildings and a large self-build vegetable garden in a walking distance may explain the high percentage of long talk/chat and joint activities in Kien Hung.

Further findings from in-depth interviews show that the majority of respondents have a family with small children under 12 (85%) and are living at a distance from their relatives. Aware of the need for help and social support, they keep close relationships with neighbors and find friends for their children.
My family will live here for a long time, if something happens, we should count on our neighbors first.(49_man_owner_Kien Hung)
I communicate with neighbors for shared interests and caring about each other.(36_woman_owner_Kien Hung)

Regarding location choice for social interaction, circulation areas including corridors, lift/lobbies, main hall/entrance were the most popular spaces; 46% of the social interactions took place in these locations. This shows that the circulation areas are extremely important in supporting social interactions between residents. People used these spaces for different purposes of social interaction:
Children cycling or playing football in the corridors.(29_woman_owner_SDU)
Mothers/grandmothers accompanying kids for co-playing/feeding in the corridors/main hall.(50_man_owner_Kien Hung)
Once in a while men gathering, drinking and smoking in the corridors.(38_woman_owner_SDU)

Quite a similar percentage of social interactions take place at the homes (15.9%) and public open spaces (16.2%). Services within buildings/neighborhoods are used for a relatively small percentage (around 10%) of social interactions. A relatively smaller percentage of interactions took place at home in Thanh Ha compared to other neighborhoods. An explanation for this may be that people have only lived here since 2018 and therefore they do not know their neighbors that well to invite them to home or vice versa. Lakes and green open spaces are within the distance of 750 m in SDU. However, there is no linking walkway thus people have to go across busy streets with dense traffic to these sites. In addition, SDU has no yards/walkway around the building. This explains the fact that this neighborhood has the lowest share of interactions in public open spaces. The ground floor front spaces of all six buildings in Dai Thanh are occupied by multiple private shops, cafés, beauty salons, etcetera. This may explain the fact that a slightly higher share of social interactions took place in shops/services in this neighborhood.

Although common rooms are available in all four neighborhoods, they are considered to be small, poorly designed and equipped. They are therefore rarely used by residents.
If there are floor events/activities we normally use the corridors, not the common room because it’s small and inconvenient.(38_woman_owner_Kien Hung)

The study of van den Berg et al. (2015) [64] indicates that services and facilities in the neighborhood such as shops, sports facilities, and recreational areas can create opportunities for social contacts among residents. However, services at building and neighborhood level are mainly private ownership with small business such as shops, minimarts, pharmacies and beauty salons, etcetera. There are not enough spaces and facilities for sport, exercise and entertainment within a distance of 500 m, resulting in a lower number of social interactions taking place in this category.
The ground floor is blocked by shops and they also occupy almost half of the walkway.(32_woman_owner_Dai Thanh)
My husband has to go to the sport club which is about 4–5 kms away from this neighborhood. On weekend we mostly go to a for cinema, shopping, relaxation in the city center.(46_woman_owner_Kien Hung)

Overall, 83.1% of the interactions were considered to be important or somewhat important, 12.5% were not important at all and only 4.4% of the recorded social interactions were indicated as very important. The share of “not important at all” interactions is significantly higher in Thanh Ha (21.1%) compared to the other neighborhoods. This also explains why there is a relatively low share of social interactions at home in this neighborhood.

### 5.5. Spaces Used by Main Purposes

Table 4 shows the description of the spaces used for different purposes of interaction. As can be seen, a greeting and short message most likely take place in the circulation areas. The same goes for a talk and chat which frequently occur in the circulation areas (28.3%), followed by peoples’ home (26.0%). Joint activities were distributed mostly between home, neighborhoods’ shops/services and public open spaces. Regarding accompanying kids for co-playing, circulation is the second choice after public open space which account for 28.2 % and 49.6% of responses, respectively.

The circulation areas were found to be popular spaces for daily social interaction. Nevertheless, the frequent use of these areas for different types of interaction was indicated as creating social annoyance and decreasing residents’ privacy and safety. This is related to the study of Altman’s privacy theory (1977) which states that if a certain level of privacy cannot be controlled or involuntarily encounters take place, social interaction can be decreased within the neighborhood.
Children cycling or playing football (sometimes hitting the ball to the neighbor’s doors/walls) therefore make noise and even damage spaces’ property.(29_woman_owner_SDU)
Children use the corridor like a street to play.(71_man_owner_Kien Hung)
Sometimes, old women drop trash, waste and even let their children pee in the elevators or main hall.(50_man_owner_Kien Hung)
Men gathering, drinking and smoking in the corridor and leaving trash.(38_woman_owner_SDU)

Information was further sought out during on-site visit and discussion on local social media to verify the current situation of residents’ behavior for social interaction in high-rise living environment in Hanoi. Social interactions taking place in the circulation areas are not only greeting or talking, but also important community meetings. Figure 3 shows images of a “corridor party” (a,b: vnexpress, n.d.) [65] and the use of lift lobbies and corridors for different purposes (c, d, e: author, 2019). Many people are in doubt about this spontaneous activity. They consider it as inappropriate in modern life and think it has a negative impact on the safety and privacy of people living in these buildings.

### 5.6. The Relationships between Personal Characteristics and the Number of Social Interactions

Table 5 presents the relationships between some personal characteristics of the respondents and the number of social interactions people have in one day using One-way ANOVA analysis. Significant differences can be noticed between the number of social interactions and age group, employment status, length of residence, and household types.

Age is found to have a significant effect on number of social interactions which is in line with the studies of Forrest and Kearns (2001) [66], Van Den Berg (2015) [42] and Moor and Mohammadi (2019) [43]. The oldest group (55 and older) recorded the lowest number of social interactions. However, results from in-depth interviews reveal that older residents have more time at home, their social networks are narrower and do have a need for social interaction in the closed-door environment.
Residents are mainly young/working age people, they have different schedules and interests, so I do not have many opportunities to meet and communicate with neighbors.(69_woman_owner_SDU)

With regards to gender, background and education level, no significant differences were found in mean numbers of social interactions in the neighborhood. Although only significant at the 10% confidence level, employment status has been found to be related to the number of social interactions. Unemployed and free-lance workers had fewer contacts with neighbors compared to employed ones. Freelancers have different working schedules and have less contact with neighbors.
I am selling things at the market and working late every day so I do not have much time for social contact.(40_man_owner_Kien Hung)

The presence of children in the household increases the likelihood of interacting with neighbors which corresponds with the study of Ferguson and Ferguson (2016) [47] and Grannis (2009) [38].
Our kids need playmates so we communicate more with neighbors to find friends for them.(37_woman_owner_Kien Hung)
I visit my neighbor at home when accompanying my kids.(29_woman_renter_Kien Hung)

The findings from interviews also confirm the positive relationship between home ownership and number of social interactions as stated by Al-Homoud and Tassinary (2004) [44] (significant at the 10% confidence level).
Rental residents don’t really communicate with us, they only present in the special events.(49_man_owner_SDU)

In this study, CHAID (chi-squared automatic interaction detection) is used to study the relationship between the number of social interactions residents have and an array of predictor variables. The advantage of CHAID according to Kemperman et al., (2009) [67] is that it provides a multi-linear instead of a single linear test of the relation between the dependent and predictor variables. Figure 4 presents the result of the CHAID analysis: clusters of residents with significantly different numbers of social interactions explained by a number of predictor variables.

First, the results show a significant split in the number of social interactions in neighborhood 4 (Dai Thanh) compared to the other neighborhoods (*p* = 0.012). This is probably due to the longest duration of stay, the provision of walkway in the surrounding area and the diversity of services at the ground level in this neighborhood. Next, the most significant difference among residents living in the other neighborhoods is whether people are employed or unemployed, the employed (on average) have 4.06 interactions, while the unemployed have 3.67 social interactions. Subsequently, a remarkable difference can be seen among employed males at different ages. On average, employed males over 34 have 4.06 social interactions and those who are younger than 34 have 3.48 interactions. Among employed residents, females in general have 4.26 social interactions which is relatively higher than an average of 3.85 in males. Within unemployed residents, a remarkable effect of household structure was found. A low number of social interactions (3.00 on average) of unemployed residents was found in families without kids compared to a significant higher number of interactions (3.77) in families with kids. The reason for this may be that unemployment reduces people’s confidence and therefore unemployed residents have a fewer number of interactions with neighbors. However, with the presence of children in their families they are more likely to interact with neighbors, mostly to seek playmates and friendships for their kids.

## 6. Conclusions

The aim of this study was to investigate how often and where people in high-rise neighborhoods interact, and how this is affected by personal and neighborhoods characteristics. Descriptive statistics were presented based on questionnaires and interaction diary data collected from 274 respondents in four different high-rise apartment buildings and neighborhoods for low-income people in the city of Hanoi, followed by bivariate analyses and CHAID analysis. In addition, findings from the in-depth interviews were added to verify the results from social interaction diaries and questionnaires.

The findings of this study indicate that most social interactions (46%) between high-rise apartment residents take place in the circulation areas. The interactions not only include greeting and talking but also gathering and accompanying playing children. However, circulation areas are not designed for these purposes. The use of these spaces for children to play, to gather for a drink or having small talk or parties, is found to create social annoyance among residents, as these activities were found to create noise and negatively impact people’s privacy, feelings of safety and cleanliness of these shared spaces. Therefore, it is important that future designs of high-rise apartment buildings provides appropriate opportunities and spaces to accommodate actual social interaction between residents, including exchanging small talk or having a long conversation, accompanying children and gathering. The design should not only encourage social interaction but also provide a certain level of privacy and safety for residents. Another finding is that older people recorded lower number of social interactions compared to younger age residents. It is essential that high-rise apartment buildings are designed in such a way that they can create more opportunities for social interactions for a heterogenous group of residents.

The rare use of common rooms and the use of corridors for different purposes reveals the needs of spaces for interaction in the near home environment. Future high-rise apartment buildings should take into account design interventions to create opportunities and shared spaces within the buildings for social interaction between their residents. Birchall (1988) [68] and Gehl (2011) [51] confirm the relation between community size and the frequent use of shared spaces: the smaller the group the more inclined residents are to participate in communal activities. Therefore, these shared spaces should vary in scale, form and be flexible to accommodate diverse activities. Due to the dense population of high-rise apartment buildings, these shared spaces should be distributed vertically within a building to support a certain number of apartments or residents: this requires further study.

About 80% of the recorded interactions are considered to be important or somewhat important; therefore, it can be concluded that the selected purposes for social interaction of this study are relevant and social interaction still matters in the daily lives of residents. This study also confirms the interrelation between personal characteristics including age, length of residence, employment status and household composition and the number social interactions among high-rise residents. However, whether frequently interacting with neighbors has positive effects on social cohesion and attachment to the neighborhood still needs further investigation.

This study, however, is subject to some limitations. First, the sample is limited to only four high-rise neighborhoods in the city of Hanoi and does not represent the entire population. Nevertheless, the sample reflects the socio-demographics of high-rise apartment buildings for low-income people. Another limitation is that the social interaction diaries were recorded for only one day. If respondents could record their diaries for more days, including both weekend and weekdays this could give more and better insights into their social interactions and use of their living environment. Finally, although this study has pointed to the importance of neighborhood characteristics in supporting social interaction among residents, how specific spatial design interventions can stimulate social interaction between residents of high-rise apartment buildings requires further research.

All in all, this study provided empirical evidence on social interactions among residents in social and low-cost high-rise apartment buildings related to their personal and living environment characteristics and the results contribute to the growing body of empirical evidence on social effects of high-rise living environments. Results of this study show the relationship between purposes for interaction and spaces used and provide information for planners and designers to design future high-rise apartment buildings that support social interaction among their residents. Since the focus of this study is on high-rise apartment buildings for low-income people, who are recognized as having limited access to resources with more needs for social support and social interactions in the neighborhoods, the findings of this paper can be applied to the situation of social and low-cost high-rise apartments in large cities which are facing rapid urbanization and population growth, specifically developing countries’ s cities, that are responding to an increasing housing demand in recent years.

## Figures and Tables

**Figure 1 ijerph-17-04619-f001:**
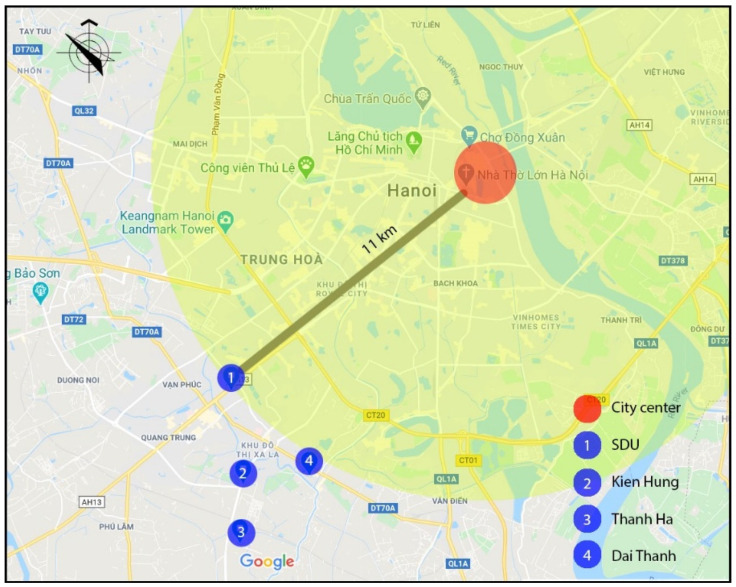
Location of the four neighborhoods. Source: the authors.

**Figure 2 ijerph-17-04619-f002:**
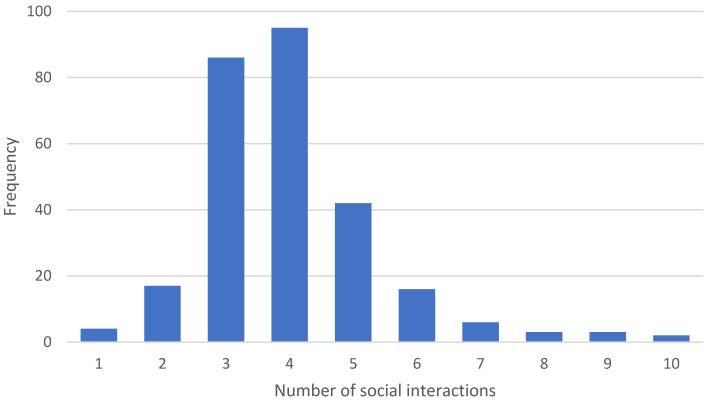
Distribution of number of social interactions in one day.

**Figure 3 ijerph-17-04619-f003:**
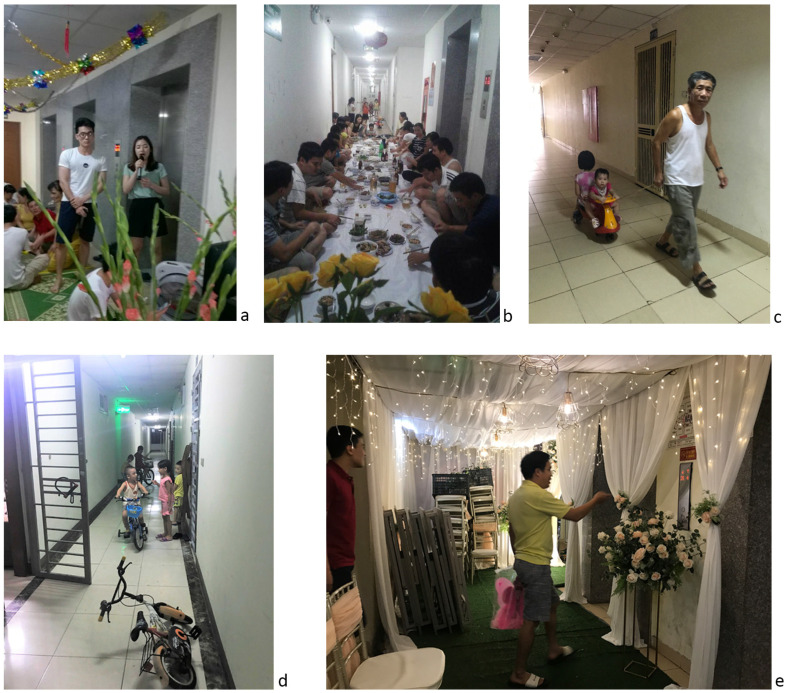
Circulation areas are occupied for different purposes of interaction. *Note:* Pictures are used with the permission of the person who revealed the face in these pictures.

**Figure 4 ijerph-17-04619-f004:**
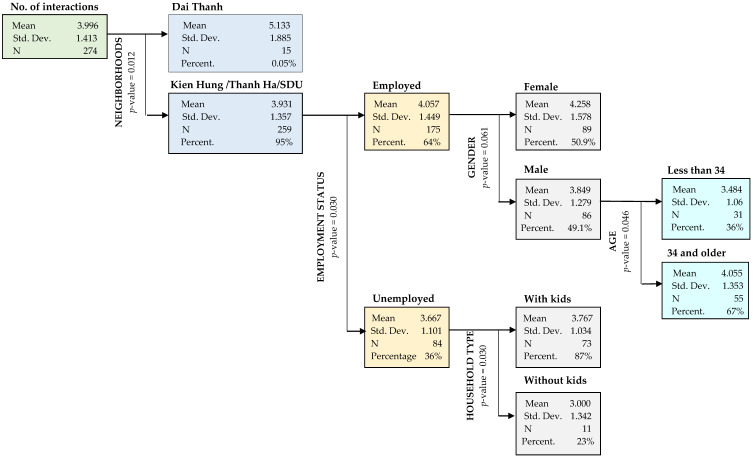
Clusters of residents based on number of social interactions.

**Table 1 ijerph-17-04619-t001:** Personal characteristics of respondents in four neighborhoods.

	Kien Hung	Thanh Ha	SDU	Dai Thanh	Total	Hanoi
	*N* = 73	*N* = 80	*N* = 106	*N* = 15	*N* = 274	
**Age**						
18–34	26.0	66.3	30.2	40.0	40.1	40.57 *
35–54	53.4	25.0	60.4	53.3	47.8	36.40 *
55 and older	20.5	8.8	9.4	6.7	12.0	23.03 *
**Gender**						
Male	45.2	48.8	46.2	46.7	46.7	49.5 *
Female	54.8	51.3	53.8	53.3	53.3	50.5 *
**Background**						
Hanoi	23.9	10.0	25.5	0	19.0	
Others	76.1	90.0	74.5	100	81.0	
**Education**						
High-school diploma or less	19.2	27.5	9.4	13.3	17.5	
Vocational training /college	9.6	13.8	13.2	33.3	13.5	
University/higher education	71.2	58.8	77.4	53.3	69.0	
**Employment**						
Employed	68.5	56.3	75.5	86.7	68.6	86.05 **
Free-lance business	6.8	27.5	10.4	6.7	14.2	
Unemployed	24.7	16.3	14.2	6.7	17.2	13.95 **
**Length of residence**						
Less than 1 year	5.5	36.3	2.8	0	13.1	
1 to 2 years	4.1	63.8	2.8	0	20.8	
2 to 5 years	30.1	0	94.3	26.7	46.0	
5 to 10 years	60.3	0	0	73.3	20.1	
**Household types**						
Without children	5.5	8.8	8.5	0	7.3	
With children	94.5	91.3	91.5	100.0	92.7	
**Ownership**						
Owner occupied	80.8	78.8	87.7	100.0	83.9	
Rental	19.2	21.3	12.3	0	16.1	

*Note:* 2019 population and housing census of Vietnam _General statistics of Vietnam. * census of Hanoi; ** census of urban area _Red River Delta. [63].

**Table 2 ijerph-17-04619-t002:** Neighborhoods characteristics.

	1. Kien Hung	2. Thanh Ha	3. SDU	4. Dai Thanh
**Housing typology**	Multiple high-rise	Multiple high-rise	Single high-rise	Multiple high-rise
**Year of occupation**	2012	2018	2015	2014
**Number of blocks**	5	3	1	6
**Block height**	19	19	36	32
**Built form**	Each building has small front court with benches	3 buildings share one front court with few benches	Building directly connected to busy street No open space, courtyard or playground	3 buildings connected to each other and surround by walkway with few benches
**Number of apartments/1 block**	288	288	512	3 blocks: 584 apartments/1 block 3 blocks: 744 apartments/1 block
**Shops/services within building**	One mini mart, one pharmacy	Mini mart, kiosks occupied the whole ground floor: small café/ restaurant/services	One small private minimart, one private cloths’ shop in ground floor	Ground floor of one building is used for market Kiosks occupied the whole ground floor: small cafés/restaurants/services
**Communal spaces**	1 common room/1 building	3 buildings share one common room	(1) Playroom (2) Study room (3) Space for exercise (4) Meeting room	1 common room/1 building
	**1. Kien Hung**		**2. Thanh Ha**	
**Neighborhood level**	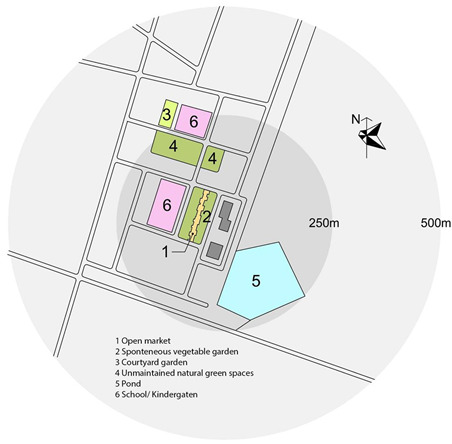		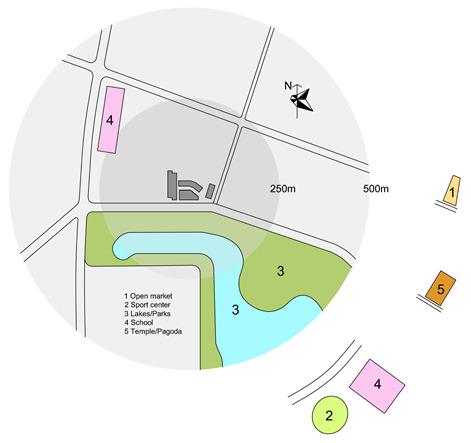	
**Green/open spaces**	No		Lake/park in the distance of 100 m	
**Sport facilities**	No		No	
	**3. SDU**		**4. Dai Thanh**	
**Neighborhood level**	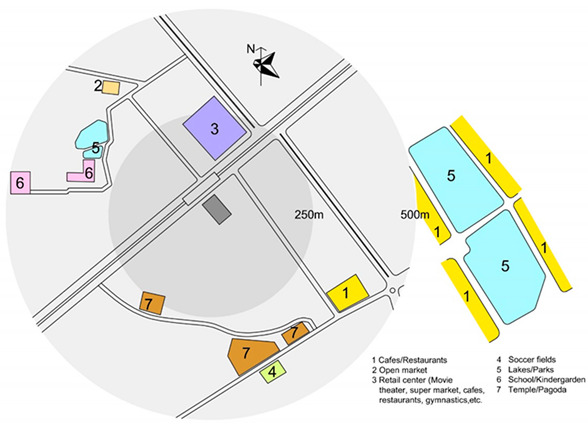		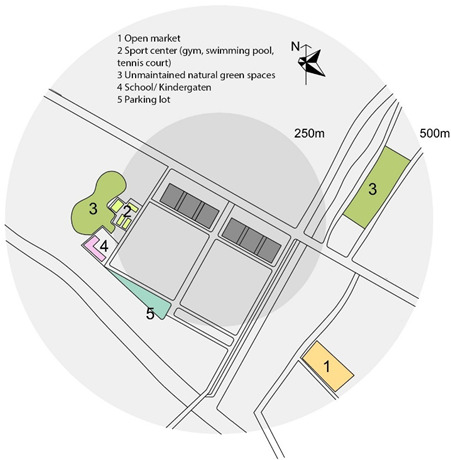	
**Green/open spaces**	Lakes/open space in the distance of 750 m		No	
**Sport facilities**	Football fields, tennis court/gym.		Gymnastics (gym, swimming pool)	

**Table 3 ijerph-17-04619-t003:** Main purposes, spaces used and importance by number of social interactions.

	**Kien Hung**	**Thanh Ha**	**SDU**	**Dai Thanh**	**Total**		
Number of respondents	73	80	106	15	274		
Average number of social interactions	4.00	3.74	4.03	5.13	4.00		
	%	%	%	%	%	**Chi-Square**	**Sig.** **(2_sided)**
**Purposes of interaction**							
Greeting/short message	32.8	46.0	53.6	42.8	45.2	62.517	0.000
Long talk/chat	37.2	33.2	30.2	31.2	33.0		
Joint activities	18.8	5.4	8.4	19.5	11.1		
Accompanying kids for co-playing	11.3	15.4	7.7	6.5	10.7		
**Spaces for interaction**							
Base home	16.4	10.4	19.7	14.3	15.9	29.977	0.003
Circulation areas *(corridor, lift/lobby, main entrance/hall)*	42.0	52.0	45.7	40.3	46.0		
Shops/service within building	11.9	9.7	11.9	10.4	11.2		
Neighborhood’s shops/service	10.6	8.4	11.9	13.0	10.7		
Public open space *(yards/playgrounds/lakes/parks)*	19.1	19.5	10.7	22.1	16.2		
**Importance of social interaction**							
Very important	6.2	2.0	4.2	7.8	4.4		
Important or somewhat important	79.1	76.8	89.2	88.3	83.1	47.652	0.000
Not important at all	14.7	21.1	6.6	3.9	12.5		

**Table 4 ijerph-17-04619-t004:** Cross-tabulation of spaces used by main purposes.

	Greeting/ Short Message	Long Talk/ Chat	Joint Activities	Accompanying Kids for Co-Playing	Chi-Square	Sig. (2-sided)
	(%)	(%)	(%)	(%)		
Base home	7.5	26.0	23.8	12.0	442.758	0.000
Circulation areas *(corridor, lift/lobby, main entrance/hall)*	73.5	28.3	4.9	28.2		
Shops/service within building	10.1	15.5	8.2	6.0		
Neighborhood’s shops/service	3.4	15.0	33.6	4.3		
Public open space *(yards/playgrounds/gardens/lakes/parks)*	5.7	15.2	29.5	49.6		

**Table 5 ijerph-17-04619-t005:** The relationships between personal characteristics and the number of social interactions.

	Mean	Number	F	Sig.
**Age**				
18–34	3.81	110	3.548	0.030
35–54	4.23	131		
55 and older	3.70	33		
**Gender**				
Male	3.90	128	1.154	0.284
Female	4.08	146		
**Background**				
Hanoi	4.27	52	2.482	0.116
Others	3.93	222		
**Education**				
High-school diploma or less	3.88	48	2.097	0.125
College or vocational training	4.43	37		
University or higher education	3.94	189		
**Employment**				
Employed	4.13	188	2.839	0.060
Unemployed	3.70	47		
Free-lance business	3.69	39		
**Length of residence**				
Less than 1 year	3.44	36	3.602	0.014
1 to 2 years	3.75	57		
2 to 5 years	4.15	126		
5 to 10 years	4.25	55		
**Household type**				
Without children	3.40	20	3.884	0.050
With children	4.04	254		
**Ownership**				
Owner occupied	4.06	230	3.009	0.084
Renter	3.66	44		

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
