# Peer review of "Where do People Interact in High-Rise Apartment Buildings? Exploring the Influence of Personal and Neighborhood Characteristics"

_ijerph, 2020, doi:10.3390/ijerph17134619_

Round 1
Reviewer 1 Report
Review of manuscript: Where do people interact in high-rise apartment buildings? Exploring the influence of personal and neighborhood characteristics Comments for authors: This is a research into the impact of personal and neighborhood characteristics on social interaction in high-rise apartment buildings, the paper has some major problems that need to be addressed. Firstly, what is the purpose of this research? There are four places that described the research’s aims differently: “This study aims to investigate the influence of personal and neighborhood characteristics on social interaction among high-rise residents.” (line 13-14), “This study makes a contribution in that it aims to investigate the influence of personal and neighborhood characteristics on social interaction among residents of high-rise apartment buildings. Specifically, the research question is: How often and where do people in high-rise neighborhoods interact, and how is this affected by personal and neighborhoods characteristics?” (line 60-63), “The aim of this study is to investigate how often and where people in high-rise neighborhoods interact, and how this is affected by personal and neighborhoods characteristics.” (line 208-209), “ The aim of this study was to investigate to what extent social interaction take place in high-rise neighborhoods and how this is affected by residents’ personal characteristics and their living environment.” (line 516-518). The aim of the research should be achieved by the research results. Secondly, in “Data collection” section, the paper describe that “the data was collected between July and September 2019 in four different high-rise apartment buildings and neighborhoods for low-income people in Hanoi, the capital city of Vietnam.” What is the meaning of “low-income” for this research? The results didn’t show its relevance with the research. If the data were collected from people with low-income, the results cannot support how social interactions are affected by personal characteristics in general. Thirdly, although authors explained its limitation of one-day diary, it is unconvinced that using one day diary’s records to conclude the general number of social interactions. Besides, results from the research lack novelty, for example, “the oldest group (55 and older) recorded the lowest number of social interactions.” It is kind of common knowledge. Finally, what methods did authors used for analyzing the data? What themes did authors get from the qualitative results? There also need to be more focused written.

Reviewer 2 Report
This study aims to investigate the extent to which social interaction takes place in high rise buildings and how this is influenced by residents' personal characteristics and the environment of the buildings. This study employs a mixed-method approach and the findings hold significance for architects. Hence, I believe this paper is relevant to the journal and its audience. Especially, the findings represented by Figure 2 is really interesting. Overall, this paper can be published after a few minor revisions, as described below:
1.In the Introduction, when you talk about mixed methods, you also need to include "physical characteristics of the neighborhood are objectively measured" since this constitutes the quantitative part of the study.
2. Under section 2.1, the last sentence of the 1st paragraph is repeated in the beginning of the next paragraph. This should be revised.
3. Why did you study 500 meter buffers of the buildings? You need to justify the selection of your threshold.
4. It would be beneficial and more legible if you present a map on which the 4 neighborhoods are located and their distance form the city center. And how are the neighborhoods seleted? Any criteria applied?
5. Why were 60 people targeted initially for the in-depth interviews?
6. On Table 2, under Dai Thanh, what are the numbers indicated as "CT8_ABC" and "CT10_ABC"? Also, what the numbers 7 and 4 at the edge of the buffers for Dai Thanh? These are figures and not a Table.
7. There are a few grammatical errors, so the manuscript should be proof read.
Reviewer 3 Report
Dear editor,
I have reviewed the manuscript entitled “Where do people interact in high-rise apartment buildings? Exploring the influence of personal and neighborhood characteristics.”
The paper has a potential, if the following comments are properly addressed in the revised version.
- In order to strengthen the scientific background of the paper further references to landmark studies should be added. The authors can add more literature review on literatures from Western and Asian countries empirically researching the factors supporting social interaction in high-rise living environment. Also, the authors can supplement previous studies that address the specific situation of the high-rise apartment residential environment in Hanoi.
- The authors can add more explanation about the reasons for the study focused on low-income areas in Hanoi and the reasons why the four apartments (Kien Hung, SDU, Thanh Ha, Dai Thanh) in Hanoi were selected as the subject of the study. In what aspects, do the authors think that those four apartments could be a representative sample of high-rise apartment in Hanoi? If those four apartments are representative samples to analyze the effects of different housing environments on social interaction, the authors might need a strategy to categorize them by including more number of samples of high-rise apartments in Hanoi.
- Another issue is the transferability of results; that is, whether the findings are applicable beyond Hanoi? The authors should make clear whether or not the results are expected to be transferable. The authors should be clear which of their findings they expect to be specific to Hanoi versus which they expect to be transferable to other cities (and if so, how similar to Hanoi does a city have to be for the findings to apply)? Conclusions such as “this study confirms the interrelation between personal and neighborhood characteristics and the number social interactions among high-rise residents.” are likely to be over-general.
- CHAID analysis was used to explore the impact of personal and neighborhood characteristics on social interaction. In CHAID Tree (line 496~514), please include % that represents sample distribution. Each branch should have enough number of sample size for reliable analysis. (e.g., branch for “Dai Thanh”, and “Without Kids” has number of samples 15 and 11, respectively.. It seems there is a limit to producing robust results.)
Round 2
Reviewer 3 Report
You have adressed the issues raised to some extend. So I won't oppose the publication of this paper.